# Ibero-American Consensus for the Management of Liver Metastases of Soft Tissue Sarcoma: Updated Review and Clinical Recommendations

**DOI:** 10.3390/cancers17081295

**Published:** 2025-04-11

**Authors:** Raquel Lopes-Brás, Paula Muñoz, Eduardo Netto, Juan Ángel Fernández, Mario Serradilla-Martín, Pablo Lozano, Miguel Esperança-Martins, Gerardo Blanco-Fernández, José Antonio González-López, Francisco Cristóbal Muñoz-Casares, Isabel Fernandes, José Manuel Asencio-Pascual, Hugo Vasques

**Affiliations:** 1Department of Medical Oncology, Hospital Santa Maria, Centro Hospitalar Universitário Lisboa Norte, 1649-028 Lisbon, Portugalmiguelmemartins@campus.ul.pt (M.E.-M.); 2Department of General Surgery and Surgical Oncology, Hospital Quironsalud Torrevieja, 03184 Torrevieja, Spain; paumozmoz@gmail.com; 3Department of Radiation Oncology, Instituto Português de Oncologia de Lisboa, 1099-023 Lisbon, Portugal; 4Department of General Surgery, Hospital Universitario Los Arcos del Mar Menor, 30739 Murcia, Spain; jaferher@outlook.com; 5Department of Surgery, School of Medicine, Catholic University of Murcia, Campus Los Jerónimos, Guadalupe, 30107 Murcia, Spain; 6Department of Surgery, Hospital Universitario Virgen de Las Nieves, Instituto de Investigación Biosanitaria Ibs.GRANADA, School of Medicine, University of Granada, 18014 Granada, Spain; mserradilla@ugr.es; 7Department of Surgical Oncology, Hospital General Universitario Gregorio Marañon, Universidad Complutense de Madrid, 28009 Madrid, Spain; 8Gulbenkian Institute for Molecular Medicine, 1649-028 Lisboa, Portugal; 9Department of Hepato-pancreatic-biliary Surgery and Liver Transplantation, Hospital Universitario de Badajoz, Universidad de Extremadura, 06080 Badajoz, Spain; 10Facultad de Medicina y Ciencias de la Salud, Universidad de Extremadura, 06006 Badajoz, Spain; 11Instituto Universitario de Investigación Biosanitaria de Extremadura (INUBE), 06080 Badajoz, Spain; 12Department of Surgery, Hospital Universitario Santa Creu i Sant Pau, 08025 Barcelona, Spain; jgonzalezl@santpau.cat; 13Peritoneal Carcinomatosis and Retroperitoneal Sarcomas Unit, San Juan de Dios Hospital, 14012 Córdoba, Spain; 14EpiDoC Unit, CHRC, Nova Medical School, Universidade NOVA de Lisboa, 1169-056 Lisbon, Portugal; 15Department of Hematology and Oncology, CUF Oncologia, 1998-018 Lisbon, Portugal; 16Hepatobiliary Surgery and Liver Transplant Unit, Gregorio Marañón University Hospital, 28007 Madrid, Spain; 17Department of Surgery, Faculty of Medicine, Universidad Complutense of Madrid, 28040 Madrid, Spain; 18Department of Surgery, Instituto Português de Oncologia de Lisboa, 1099-023 Lisbon, Portugal; hugovasques@sapo.pt

**Keywords:** soft tissue sarcoma, liver metastases, oligometastatic disease, retroperitoneal sarcoma, ablative techniques, liver surgery, ablative techniques

## Abstract

Liver metastases from soft tissue sarcoma are very rare. The standard treatment for metastatic sarcoma is primarily based on chemotherapy, often with poor results. However, when metastases are confined to the liver and are amenable to surgery or ablative techniques, radical treatment could be an option, keeping patients disease-free and without the need for systemic cytotoxic treatment for long periods of time, with a positive impact on overall survival. A multidisciplinary working group of experts in sarcoma and liver surgery reviewed the literature and available evidence and developed a set of clinical recommendations to be voted and discussed in the I Ibero-American Consensus on the Management of Metastatic Sarcoma, held during the III Spanish-Portuguese Update Meeting on the Treatment of Sarcomas in May 2024. Herein, the voting results of this meeting and the resulting consensus recommendations are presented, and their applicability, strengths, and limitations are discussed.

## 1. Introduction

Current evidence indicates that approximately half of patients with soft tissue sarcoma (STS) are diagnosed with or will develop distant metastases during the course of their disease, with an overall survival (OS) of less than 20% [1,2].

Metastatic STS (mSTS) is primarily treated with systemic agents, namely conventional chemotherapy, in contrast to localized disease where surgery is the mainstay of treatment [3,4]. The most common site of metastatic disease in STS is the lung, with liver-only metastases being uncommon. Patients with retroperitoneal and intra-abdominal/visceral STS are more likely to develop liver-only metastatic disease than patients with extra-abdominal primary STS. This is particularly evident in the setting of metastatic retroperitoneal sarcoma (RPS).

Patients with liver metastases from STS have a dismal prognosis, and optimal management has not been established. In a retrospective cohort study by Okamoto et al. [5], the risk of developing liver metastases was significantly higher in RPS and leiomyosarcoma (LMS) compared to other STSs. The 2-year survival rate after liver metastases was 21.7%.

The standard treatment for localized primary RPS is surgical resection [3,4]. Given the lack of significant benefit from systemic therapies, this is also true for large, aggressive RPS histologies with a high risk of recurrence and for metastatic RPS.

Although efforts have been made to develop treatment guidelines for metastatic RPS, these have been hampered by the paucity of studies evaluating the role of surgery in this setting [6]. In addition, surgical resection is increasingly performed for liver metastases from colorectal and neuroendocrine tumors, but remains controversial for liver metastases from STS, a much rarer disease with uncertain and often poor prognosis.

Hepatic metastases from STS are most commonly treated non-surgically, with systemic therapy (more commonly conventional chemotherapy) remaining the primary treatment with palliative intent but hoping to achieve longer disease-free interval (DFI) and treatment-free interval (TFI). Currently, there is no consensus on the role of resection for liver metastases from STS or its optimal timing in relation to systemic treatment and/or treatment of the primary tumor, and no prospective data to guide the therapeutic strategy for these patients. In addition, most studies of hepatectomy for sarcoma metastases are retrospective and have included small numbers of patients and/or a heterogeneous group of histologies (gastrointestinal stromal tumors [GISTs] with non-GIST sarcoma or other STS histologies). GISTs with liver metastases were excluded from this consensus because the development of tyrosine kinase inhibitors has completely changed the management, course, and prognosis of metastatic GISTs.

Longer OS has been achieved in selected sarcoma patients who received multimodality treatment including surgery for metastatic disease [5,7,8]. Careful patient selection for surgical resection of liver metastases from STS may prolong survival, particularly in patients with good performance status, longer DFI, oligometastatic disease (OMD), and resectable gross hepatic disease. In fact, for patients with limited metastatic burden, surgical resection of all metastatic sites can be recommended because long-term remission can be achieved [3,4]. Regarding the management of the primary tumor, both the National Comprehensive Cancer Network (NCCN) and the European Society for Medical Oncology (ESMO) recommend a localized approach when there are limited synchronous metastases and radical treatment (surgery and/or ablative techniques) is feasible.

It is widely accepted that patients with mSTS should be evaluated and managed by a multidisciplinary team of medical, radiation, and surgical oncologists specializing in STS management. Given the rarity of liver metastases from sarcoma, the heterogeneity of histological subtypes, and the lack of randomized trials evaluating the role of surgery, local ablative treatments, or optimal multimodal treatment strategies, the Spanish and Portuguese Sarcoma Groups recognized the need for an international, multicenter, multidisciplinary discussion on this topic. In 2024, these two groups held a consensus conference on the management of liver metastases from sarcoma. The meeting brought together a multidisciplinary panel of surgeons, medical oncologists, and radiation oncologists to discuss the diagnosis, treatment, and follow-up of patients with liver metastases from STS. The goal was to develop a set of recommendations informed by the best available evidence and expert opinion, focusing on controversial issues not adequately addressed in current guidelines. This initiative aims to contribute to the existing body of literature and provide a much-needed resource for clinicians involved in the management of sarcoma patients worldwide.

## 2. Methods

On 10 May 2024, the III Spanish-Portuguese Update Meeting on the Treatment of Sarcomas was held at Gregorio Marañon University Hospital in Madrid, Spain. The meeting was co-organized by the Sarcoma Group (Sección de Tumores Mesenquimales y Sarcomas) of the Spanish Association of Surgeons (Asociación Española de Cirujanos, AEC) and the Portuguese Sarcoma Study Group (Grupo Português de Estudos de Sarcomas, GPES). As part of the event, the I Ibero-American Consensus Conference on the Management of Metastatic Sarcoma was convened. The aim of this conference was to address and discuss controversial issues regarding the management of patients with metastatic sarcoma. The conference included a multidisciplinary panel of leading experts from Portugal, Spain, and Latin American countries and was co-chaired by Dr. J.M. Asencio and Dr. H. Vasques. Two expert working groups were established, each focusing on a specific area, with the following appointed chairs:Management of peritoneal sarcomatosis–Chair: Dr. F.C. Muñoz-CasaresManagement of liver metastases from STS–Chair: Dr. I. Fernandes

The planning, preparation, and execution of the consensus conference was carried out over a 3-month period prior to the meeting. A narrative and thorough literature review was conducted through an in-depth search of the MEDLINE database via the PubMed^®^ platform for publications on oligometastatic disease, surgical management of hepatic metastases (from sarcoma or mixed histologies), and systemic treatment strategies for STS in the context of resected metastatic disease. No restrictions were applied regarding the publication date. The working group was composed of 13 clinicians, including 3 medical oncologists and one radiation oncologist specializing in the treatment of soft tissue and bone sarcomas, as well as 9 surgeons specializing in sarcoma and liver surgery. All recommendations formulated by the group were supported by an assigned level of evidence and strength of recommendation according to the ‘Infectious Diseases Society of America-United States Public Health Service Grading System’ [9].

During the meeting, participants used the Slido interactive platform to vote by scanning a QR code with their mobile devices. Consensus was defined as agreement by at least 80% of the voters, categorized as follows: weak consensus: 80–89% agreement; strong consensus: 90–99% agreement; unanimous consensus: 100% agreement. Recommendations that did not achieve consensus during the meeting are presented and discussed here.

This article details the results of Working Group 2 of the Consensus Conference, including all recommendations and a summary of supporting evidence. The final manuscript was reviewed and approved by all working group members.

## 3. Results

### 3.1. What Is the Definition of Oligometastatic Disease for Sarcoma?

Discussing the role of local approaches for liver metastases from STS is most relevant in the setting of low-burden disease, as the elimination of all detectable disease could lead to long-term remission. Frequently, OMD is generically established as a limited number of clinically detectable metastases, normally one to five lesions, located in one or a limited number of organs [10]. Following the definition of Hellman and Weichselbaum, OMD can be defined as an “intermediate state between localized cancer and wide-spread metastatic disease, where radical local treatment of the primary cancer and of all metastatic lesions might have a curative potential” [10]. This definition, more than considering the size and number of lesions, puts emphasis on the outcome. OMD is considered as an intermediate state where local or treated metastases control may improve systemic disease control, including the potential to delay (or avoid) the systemic spreading of the disease and the use of systemic chemotherapy.

Other definitions have been proposed by different oncology societies. In a consensus publication by EORTC and ESTRO [11], OMD is defined as an overall umbrella term that is further subcategorized in nine clinical states defined by disease burden and clinical history (disease evolution in time). In this definition, there is no specific number of lesions and affected organs, as the authors consider biologic and clinical behavior the main factors to characterize the OMD as a specific clinical state, and that this classification could better help guide treatment decisions.

While several definitions and considerations about OMD exist, they mostly concern carcinomas (e.g., ESMO definition [2]), or even, a specific anatomical setting, like colon cancer or non-small cell lung cancer [12]. The more traditional clinical definitions using specific numbers of lesions and metastatic sites follow radiological appearances. However, the biology of the disease and molecular characteristics with prognostic and/or treatment implications may have a role in defining OMD and whether this concept really exists in some types of cancer. Defining OMD is a challenge as it probably is a dynamic entity whose definitions and clinical relevance are dependent on primary tumor site, tumor biologic behavior, and extent of metastatic burden and metastatic sites. 

Regarding STS, the lung is the most common site of metastases in most histologies and other metastatic sites are usually associated with worse prognosis (e.g., liver, central nervous system [CNS]). Different histological subtypes are also associated with different prognosis, and this should also be considered, when deciding to treat a patient as oligometastatic. As no definition exists for STS OMD, we propose the definition of five or fewer lesions and two or fewer affected organs, considering that the most common site of metastization is the lung (even for intra-abdominal sarcomas) and that in patients with liver metastases from STS, the presence of concurrent extra-pulmonary extra-hepatic lesions could mean a more systemic disease and portend a dismal prognosis, making it less likely that the patient could be benefit from focal therapy.

Appropriate staging of metastatic disease is imperative in defining OMD. Confirmation of oligometastatic state should include morphological imaging and additional modalities such as ^18^F-fluorodeoxyglucose positron emission tomography (^18^F-FDG PET)- computed tomography (CT) and brain magnetic resonance imaging (MRI) (see below Section 3.4. How should sarcoma liver metastases be radiologically evaluated?) [2,3]. To be considered OMD, all metastatic sites must be safely treatable by local ablative treatment modalities (surgical and non-surgical). The involvement of a diffuse organ such as malignant pleural, pericardial, peritoneal or leptomeningeal carcinomatosis usually excludes patients from being considered as oligometastatic in the setting of carcinomas. Similarly, it could also be the case for STS, although, for example, in selected cases, peritoneal disease may be amenable to radical therapy [13].

Lastly, the control of primary tumor is also a matter of concern, as this is also related to the timing of oligometastases—synchronous OMD or oligorecurrence. In the latter, the primary tumor is usually already treated (with surgery most often) but in the former, primary tumor needs to be addressed too, as the ultimate goal of radical treatment in OMD is to render the patient disease-free, hoping to prolong survival. In this scenario, our consensus opinion is that the primary tumor should be treated before or at the same time as the metastasectomy.

We must emphasize that any treatment focused in OMD must have as the goal to cure the disease having DFS as an endpoint for outcome evaluation or delaying the need for palliative systemic treatment.

**Recommendation 1.1.** OMD in STS should be defined by mSTS with a number of metastatic lesions ≤5 in ≤2 different metastatic sites, confirmed by morphologic staging and additional imaging and/or nuclear medicine modalities, as appropriate, in which the primary and all metastatic sites are amenable to radical treatment (either surgical and/or nonsurgical ablative procedures).

Level of evidence: V

Strength of recommendation: B

Level of consensus: 92% (33) yes, 8% (3) no (36 voters)

**Recommendation 1.2.** Involvement of a diffuse organ such as malignant pleural, pericardial, peritoneal or leptomeningeal sarcomatosis should exclude patients from being considered as oligometastatic, as this scenario would render the disease ineligible for radical treatment.

Level of evidence: V

Strength of recommendation: B

Level of consensus: 88% (30) yes, 12% (4) no (34 voters)

### 3.2. What Is the Definition of Synchronous and Metachronous Oligometastatic Disease?

The time of onset of OMD should be taken into account when deciding on the treatment strategy, namely the use of systemic treatment and/or local techniques and the sequencing of these different treatments, as a short previous recurrence-free interval may indicate a poorer prognosis [3]. Synchronous OMD is defined by the presence of oligometastatic lesions at the initial diagnosis of the primary tumor or within a few months thereafter. Although no universally accepted definition exists, a commonly used criterion defines synchronous disease as the detection of oligometastases within six months of the primary tumor diagnosis [14]. In contrast, metachronous OMD, often referred to as oligo-recurrence, develops at least six months after the initial diagnosis.

**Recommendation 2.1.** Synchronous OMD is defined as OMD detected either at the initial diagnosis of the primary tumor or within six months thereafter, with a limited number of metastases detected simultaneously, with a disease burden of ≤5 metastatic lesions in ≤2 different metastatic sites.

Level of evidence: V

Strength of recommendation: B

Level of consensus: 83% (30) yes, 17% (6) no (36 voters)

**Recommendation 2.2.** Metachronous OMD is defined as OMD detected more than 6 months after the initial diagnosis, with a limited number of metastases detected simultaneously, with a disease burden of ≤5 metastatic lesions in ≤2 different metastatic sites.

Level of evidence: V

Strength of recommendation: B

Level of consensus: 81% (30) yes, 19% (7) no (37 voters)

### 3.3. What Is the Role of Biopsy in Sarcoma Liver Metastases?

The role of biopsy is reasonably well established at diagnosis of primary tumor from STS. However, the role of biopsy in STS metastases is not consensual.

As for primary STS, the preferred technique for biopsy of liver metastases is image-guided percutaneous core needle biopsy (CNB) over fine needle aspiration cytology (FNAC), since the amount of tissue obtained with CNB allows for accurate determination of histology and nuclear grade [15]. Furthermore, it is the only technique that allows molecular studies, such as next-generation sequencing (NGS), which can be very important in the context of metastatic disease. In general, liver biopsy is considered safe, but there is a risk of complications that can be life-threatening or delay the initiation of treatment. The main complications include bleeding, in the form of hepatic hematoma or hemoperitoneum, occurring in up to 10% of patients, as well as organ perforation, sepsis, and, in some cases, death [16].

It is important to note that, the timing of appearance of metastatic disease and the planned treatment strategy may dictate the need for a biopsy of a metastatic lesion. In synchronous OMD, radiologically compatible lesions with metastases, a biopsy could be waived [17]. On the other hand, however, if induction systemic treatment is planned, as could be often the case in this setting, a biopsy could provide a diagnostic confirmation. In metachronous OMD, the concern for other histologies like liver metastases from carcinoma (e.g., colon cancer) or neuroendocrine tumors or primary liver cancer (e.g., hepatocellular carcinoma) should be considered before planning a metastasectomy. In retrospective series regarding liver metastasectomy, pre-treatment biopsy of liver metastases was not required regardless of metachronous or synchronous disease [17] or, most often was not mentioned in the report [5,18,19,20,21], emphasizing the lack of data on this topic.

**Recommendation 3.1.** In resectable metachronous liver metastases, biopsy should not be mandatory but could be considered if the tumor origin is uncertain (e.g., long DFI, risk factors for hepatocellular carcinoma, metachronous colorectal cancer) or if a preoperative systemic treatment approach is proposed.

Level of evidence: V

Strength of recommendation: B

Level of consensus: 97% (36) yes, 3% (1) no (37 voters)

**Recommendation 3.2.** If synchronous OMD and induction systemic treatment is planned, biopsy should be performed prior to treatment initiation.

Level of evidence:

Strength of recommendation:

Level of consensus: 62% (23) yes, 38% (14) no (37 voters)

**Recommendation 3.3.** A minimum of four large-gauge cores (14–16 G) is recommended to ensure adequate tissue sampling.

Level of evidence: IV

Strength of recommendation: A

Level of consensus: 89% (32) yes, 11% (4) no (36 voters)

### 3.4. How Should Sarcoma Liver Metastases Be Radiologically Evaluated?

Imaging plays a central role in assessing patients with liver metastases. Ultrasound, MRI, or computed tomography (CT) scans are commonly used to characterize liver lesions anatomically and assess resectability, as well as estimate the future liver remnant (FLR). Contrast-enhanced CT remains a widely available and fast imaging modality that provides good anatomical detail and is particularly useful for assessing vascular involvement. However, its sensitivity for detecting small liver lesions (<1 cm) is lower compared to MRI, with reported sensitivity and specificity of approximately 70–85% and 85–90%, respectively. In contrast, MRI, especially when using liver-specific contrast agents, offers superior soft tissue contrast and is the preferred modality for characterizing small lesions and identifying additional metastases beyond those detected by CT. This makes MRI particularly valuable for surgical planning, with a reported sensitivity of 85–95% and specificity of 90–95% [22].

^18^F-FDG PET is a valuable tool in metabolic imaging for sarcomas, namely in detecting occult metastasis and early metabolic changes following treatment. This is particularly true histological subtypes that display higher avidity for glucose, e.g., osteosarcoma, chondrosarcoma, Ewing’s sarcoma, and rhabdomyosarcoma, and also for histologies that are prone to lymph node metastases, like epithelioid sarcoma, clear-cell sarcoma, synovial sarcoma and angiosarcoma [23]. ^18^F-FDG PET can also be useful in radiologically unspecific liver lesions to confirm or rule out malignant potential. However, its sensitivity varies depending on the histological subtype, with lower detection rates for sarcomas that exhibit low FDG avidity. Reported sensitivity for PET/CT ranges from 60 to 80%, with specificity between 85 and 95%. Although there is no consensus in the literature regarding the need of ^18^F-FDG-PET/CT for all patients with metastatic disease, it should be considered in all patients that are candidates to metastatic resection to confirm the oligometastatic stage of the disease or after upfront/induction systemic treatment to evaluate response and rule out extra-hepatic progression. In a recent study [24], the use of ^18^F-FDG PET/CT scan increased the rate of additional metastases’ detection in STS and bone sarcomas that were being considered for salvage therapy, impacting the clinical management of a third of the patients.

Brain MRI has been suggested to rule out the possibility of brain metastases in suitable subtypes such as those that tend to concomitantly metastasize to the CNS—alveolar soft part sarcoma, clear cell sarcoma, angiosarcoma. An MRI of the total spine is also recommended for myxoid liposarcoma, alveolar soft part sarcoma and angiosarcoma [5].

**Recommendation 4.1.** Iodinated contrast-enhanced CT scan is recommended to evaluate vascular involvement and the abdominal cavity (especially in RPS).

Level of evidence: V

Strength of recommendation: B

Level of consensus: 100% (36) yes, 0% (0) no (36 voters)

**Recommendation 4.2.** Hepatic MRI is recommended for diagnosis and evaluation of lesions smaller than 1 cm and for characterization of lesions of uncertain origin..

Level of evidence: V

Strength of recommendation: B

Level of consensus: 100% (36) yes, 0% (0) no (36 voters)

**Recommendation 4.3.** ^18^F-FDG-PET/CT is recommended to exclude extrahepatic metastases in appropriate subtypes (e.g., lymph node metastases).

Level of evidence: V

Strength of recommendation: B

Level of consensus: 84% (32) yes, 16% (6) no (38 voters)

### 3.5. In Which Patients with Metastatic Soft Tissue Sarcoma Is Hepatic Metastasectomy Indicated?

There is increasing evidence that patients with lung metastases from sarcoma have more favorable outcomes compared to metastases at other sites, thus justifying the need for aggressive treatment [25,26]. However, for patients with non-lung metastases, there is little evidence of survival benefit from metastatic resection, and any crosstalk from the lung evidence must be interpreted with caution.

Regarding liver metastases from non-GIST sarcomas, the surgical indications and prognosis of surgical and non-surgical ablative treatments are not as well understood as in colorectal cancer or neuroendocrine tumors [27,28]. Many of the studies on surgical treatment of sarcoma liver metastases do not differentiate between GIST and non-GIST sarcoma metastases, or even between different histologies grouped together as ‘non-colorectal liver metastases’ [29], making it difficult to extrapolate the benefit of ablative treatments to non-GIST sarcoma liver disease. Despite the heterogeneity of the different sarcoma histotypes and their behavior, some series have reported that surgical resection for single or oligometastatic liver disease may be comparatively as effective as lung metastasectomy and therefore may be associated with long-term survival benefit in selected cases [18,19,30].

Although there are no randomized trials on mSTS in the liver, some conclusions can carefully be extrapolated from observational studies such as METASARC [31], which analyzed real-world evidence of sarcoma metastases management to evaluate the treatment modalities of patients with mSTS and their outcomes according to histological subtype, treatment modality, and various prognostic factors. This study states that locoregional treatment of metastases should always be included in the therapeutic strategy when feasible. Of the 2225 mSTS patients studied between 1999 and 2013 in the French Sarcoma Group, 49% received locoregional treatment (LRT): radiotherapy (RT) in 254 patients, radiofrequency ablation (RFA) in 42, and surgery (with or without RT/RFA in 320. Nineteen percent (410) of patients treated with LRT therapies had liver metastases, and 224 patients were alive 5 years after diagnosis of metastases, 83.48% of whom had received some LRT for metastatic disease. These results are consistent with the multivariate analysis, which showed that the most significant factor associated with a higher probability of 5-year survival was LRT (odds ratio [OR] 7.41; 95% confidence interval [CI] 4.42–12.41).

For the purpose of elaborating this consensus, the authors followed a narrative review of the most recent series of resected liver metastases from non-GIST soft tissue sarcomas, which are summarized in Table 1. Despite the small sample, consistent with the rarity of the disease, a survival advantage can be observed in resected patients. The EORTC historical series published in 1999 described a poor 1- and 2-year OS in the chemotherapy alone group of 42% and 13%, respectively [32]. Goumard et al. also described worse 1-, 3-, and 5-year OS rates of 50%, 13% and 4% in unresected patients [18]. Lochner found a median OS after first documentation of metastatic disease of 24 months (95% CI 21–33) for mSTS patients [33]. In this series, the 1-, 2-, and 5-year OS rates were 70.0% (95% CI 64–77), 49.9% (95% CI 43–58), and 24.8% (95% CI 19–33), respectively. Regarding the seven studies included in this review (Table 1), the median 5-year OS after liver metastases resection was 43%, which is an exceptionally high OS rate. Despite the prognostic advantage that local treatment of metastases may endorse, the favorable outcomes of the reviewed series may be the result of including well-selected patients and treatment in sarcoma reference centers.

Some small series have analyzed the prognosis of liver metastasectomy, defining three distinct groups: GIST, LMS, and other sarcomas [18,20]. In these studies, LMS seemed to have better outcomes after liver metastasectomy among sarcomas metastasizing to the liver, suggesting that this histologic subtype should be analyzed separately.

Some prognostic factors should be considered when evaluating a patient with mSTS with liver disease for surgery, and surgical resection should be preferred in cases with favorable prognostic features. In a systematic review of resected liver metastases of non-GIST sarcomas, Tirotta et al. found that DFI > 6 months was a good prognostic factor associated with survival after hepatic resection, while R2 resection, size >10 cm, and presence of extrahepatic disease were associated with decreased survival [19]. Similarly, Goumard et al. found that the only factors associated with survival were larger metastatic size (>10 cm), extrahepatic metastases, and DFI < 6 months, which was also associated with recurrence-free survival (RFS) [18]. In this context, and considering the factors associated with improved survival after STS resection of lung metastases (small size < 2 cm, long disease-free survival [>12–18 months] between primary tumor resection and metastatic recurrence, and limited number of nodules [<3–5]) [26], we recommend surgical resection of liver metastases in oligometastatic (cf. Section 3.1) and metachronous (cf. Section 3.2) disease when complete resection is feasible. Major hepatectomies, such as right or left hepatectomy, should not be a formal contraindication to surgery, as they may dependent on anatomical factors and are not directly related to a higher burden of metastatic disease, nor should bilobar spread if there are few metastatic nodules that may allow parenchymal sparing surgery of different segments. However, in cases of bilobar spread requiring complex resections (e.g., vascular involvement), surgery should be discouraged because the expected risk-benefit ratio is likely to be unfavorable.

The panel acknowledges that it is not possible to cover all the different clinical scenarios of liver metastases from sarcoma, and that the decision for surgical treatment may be at the discretion of the treating physician, based on a case-by-case basis, best expert opinion, and multidisciplinary board discussion in reference centers for both sarcoma and hepato-pancreato-biliary (HPB) surgery.

**Recommendation 5.1.** Surgical resection of liver metastases should always be previously discussed within a multidisciplinary tumor board at a sarcoma referral or network center. It is important to weigh the risk of surgery against the expected benefits.

Level of evidence: V

Strength of recommendation: B

Level of consensus: 100% (38) yes, 0% (0) no (38 voters)

**Recommendation 5.2.** Surgical resection of liver metastases should only be performed in reference or specialized high-volume centers with experience in liver surgery.

Level of evidence: V

Strength of recommendation: B

Level of consensus: 97% (34) yes, 3% (1) no (35 voters)

**Recommendation 5.3.** For metachronous disease, liver resection should be considered in patients diagnosed with STS and oligometastatic disease confined to the liver and amenable to complete surgical resection.

Level of evidence: V

Strength of recommendation: B

Level of consensus: 100% (36) yes, 0% (0) no (36 voters)

**Recommendation 5.4.** In highly selected patients with metachronous OMD with a favorable prognostic profile that includes liver metastases and limited extrahepatic disease (lung only), surgical resection may also be considered if all lesions are potentially resectable.

Level of evidence: V

Strength of recommendation: C

Level of consensus: 92% (34) yes, 8% (3) no (37 voters)

**Recommendation 5.5.** Liver resection of OMD from STS should be discouraged in patients presenting with bilobar liver metastases requiring complex hepatectomies (e.g., with vascular involvement or central location).

Level of evidence: V

Strength of recommendation: D

Level of consensus: 57% (20) yes, 43% (15) no (35 voters)

### 3.6. What Is the Best Surgical Approach?

The goal of surgical resection of sarcoma liver metastases is to achieve R0 resection while preserving adequate function of the remaining organ. R0 resection is defined as microscopically negative margins, and preoperative assessment of resectability can be safely evaluated by HPB surgeons or surgical oncologists [36].

The surgeon will adapt the technique and extent of resections to the goal of R0 resection, favoring liver-sparing resections whenever possible [37], while major hepatectomy may be warranted in selected cases with good prognostic features. Each surgical procedure should be individualized based on patient characteristics, i.e., disease extension, DFI, previous treatment with hepatotoxic chemotherapy, etc.

When planning the surgical approach and extent of resection, a thorough review of previous surgical procedures through the medical record—vascular or visceral resections, previous hepatectomies, anastomoses, bowel resections—is mandatory to prevent complications and evaluate the feasibility of resection of metastases. Any iterative abdominal surgery increases morbidity and the risk of postoperative complications (cf. Section 3.9). In this regard, the number of previous abdominal surgeries should be taken into account before considering a new intervention [6].

Minimally invasive approaches (MIS), such as laparoscopic or robotic surgery, may be considered when obtaining an R0 resection of all liver disease is technically feasible—e.g., small peripheral lesions. When considering MIS, the precedent of major abdominal surgery for primary RPS (especially in metachronous disease) should be considered, with the most favorable cases being those of pelvic location or with retroperitoneal approaches to the primary tumor.

**Recommendation 6.1.** The primary goal of resection for sarcoma liver metastases should be to achieve an R0 resection.

Level of evidence: V

Strength of recommendation: B

Level of consensus: 97% (35) yes, 3% (1) no (36 voters)

**Recommendation 6.2.** The feasibility of liver surgery to achieve an R0 resection should be evaluated by an experienced HPB surgeon at a reference center for HPB pathology or surgical oncology.

Level of evidence: V

Strength of recommendation: B

Level of consensus: 97% (35) yes, 3% (1) no (36 voters)

**Recommendation 6.3.** The extent of surgery should be determined by anatomic factors (location of lesions) and oncological principles (R0 resection), favoring parenchyma-sparing resections.

Level of evidence: V

Strength of recommendation: B

Level of consensus: 100% (36) yes, 0% (0) no (36 voters)

**Recommendation 6.4.** Major hepatectomy may be justified in selected cases with good prognostic features, with individualization of the approach based on the specific characteristics of each case.

Level of evidence: V

Strength of recommendation: B

Level of consensus: 94% (34) yes, 6% (2) no (36 voters)

**Recommendation 6.5.** Minimally invasive liver resection (laparoscopic or robotic) may be considered when technically feasible, always with the goal of achieving an R0 resection.

Level of evidence: V

Strength of recommendation: B

Level of consensus: 100% (35) yes, 0% (0) no (35 voters)

### 3.7. How Should Patients Be Evaluated Before Surgery?

The extent of liver resection strongly correlates with the risk of postoperative liver failure. While this relationship may seem intuitive and easy to assess, it is actually the volume of the remaining liver (i.e., the future liver remnant, or FLR) that is more predictive of outcomes and is therefore critical to accurately assess [38]. It has long been recognized that patients with a lower FLR are at increased risk of liver dysfunction, although the exact threshold below which resection should be avoided remains controversial.

Recognizing these important differences, the optimal cutoff for patients with a normal background liver appears to be in the range of 25%. However, patients who have undergone extensive chemotherapy and have an estimated standardized FLR between 30% and 40% should be thoroughly evaluated for evidence of underlying liver dysfunction. In such cases, prophylactic liver hypertrophy techniques (e.g., portal vein embolization, hepatic vein deprivation, associating liver partition, and portal vein ligation for staged hepatectomy [ALPPS]) could be considered [39].

To optimize postoperative outcomes and reduce the risk of potential complications, patients’ preoperative evaluation should prioritize the assessment of comorbidities, performance status, nutritional status, kidney function, and the American Society of Anesthesiologists (ASA) Physical Status Classification System.

**Recommendation 7.1.** When planning a major hepatectomy, preoperative assessment of FLR should follow recommendations derived from experience in other contexts where major hepatectomy is performed.

Level of evidence: V

Strength of recommendation: B

Level of consensus: 100% (37) yes, 0% (0) no (37 voters)

**Recommendation 7.2.** Preoperative patient assessment should follow the principles of any major surgery (assessment of comorbidities, performance status, and ASA physical status).

Level of evidence: V

Strength of recommendation: B

Level of consensus: 100% (37) yes, 0% (0) no (37 voters)

### 3.8. When Should Surgical Resection of Liver Metastases Be Considered?

As previously discussed, the terms metachronous and synchronous disease refer to the DFI between the diagnosis of the primary tumor and metastatic recurrence. The time interval that distinguishes metachronous from synchronous disease is not fully standardized (cf. Section 3.2). For the purpose of discussing the timing of liver metastasectomy, synchronous disease has been assumed when a time interval ≤ 6 months from the initial diagnosis has occurred, as this is the most commonly used time interval in the studies presented here (cf. Recommendations 2.1 and 2.2).

In terms of tumor behavior and biology, synchronous mSTS is likely to have a worse prognosis than metastatic oligoprogression with longer DFI (i.e., >12–18 months), as the latter may represent a more indolent disease. From this perspective, this panel advocated a radical approach (either surgical or with ablative techniques) to liver disease in metachronous metastases, and systemic treatment as a front-line approach in synchronous OMD. González-Abós et al. proposed an extension of the indication for resection of liver metastases in selected patients with synchronous metastases who show stability of extrahepatic disease after systemic “neoadjuvant” treatment [17]. In our opinion, in a few selected cases of synchronous OMD, surgical salvage of residual liver disease can be discussed if there has been a good response to systemic treatment and if complete resection of both the primary tumor and liver metastases is feasible.

**Recommendation 8.1.** In patients with STS OMD, surgery for liver metastases could be considered for metachronous lesions (DFI ≥ 6 months) as upfront treatment or after systemic therapy (especially in subtypes with better response to chemotherapy).

Level of evidence: V

Strength of recommendation: B

Level of consensus: 94% (34) yes, 6% (2) no (36 voters)

**Recommendation 8.2.** In patients with STS OMD and synchronous disease, systemic treatment (chemotherapy, etc.) should be the preferred front-line approach.

Level of evidence: V

Strength of recommendation: B

Level of consensus: 86% (30) yes, 14% (5) no (35 voters)

**Recommendation 8.3.** In patients with STS OMD and synchronous disease, liver surgery may be considered for hepatic residual disease, if there is a good response to systemic treatment (at least stable disease) and the primary tumor is resectable.

Level of evidence: V

Strength of recommendation: B

Level of consensus: 100% (34) yes, 0% (0) no (34 voters)

### 3.9. Remetastasectomy: Is There a Role for Surgery for Liver Recurrence?

Given the paucity of evidence for liver metastasectomy in STS, studies addressing the issue of remetastasectomy for liver recurrence are even rarer. In the retrospective series by Goumard et al. [18], 96 patients (76%) had a previous surgical resection, including previous liver resection, although these data were not explored. Older studies including GIST or non-sarcoma cases included patients with previous liver resection [40,41].

A long treatment-free interval (TFI), i.e., >12 months, is the most favorable prognostic factor for survival in metastatic soft tissue sarcomas (all sites) [35,42]. In patients with a long TFI and good performance status, local treatment of liver metastases by either surgery or ablative techniques (e.g., RFA) can be considered with the goal of controlling the disease, preserving liver function, and minimizing morbidity [30].

Due to the paucity of data regarding remetastasectomy, this panel recommends a very conservative approach for liver recurrence. When considering reintervention for liver metastases, the number of previous abdominal surgeries should be considered [6]. As explained above (cf. Section 3.6), with each subsequent abdominal surgery, the risk of morbidity and postoperative complications increases and the chance of success (R0 resection) may decrease.

The panel supports the use of preoperative systemic treatment in recurrent liver disease if no prior chemotherapy has been administrated, in histologies with high chemosensitivity such as LMS, and in relapses with a short DFI. Treatment decisions should be made by a multidisciplinary tumor board on a case-by-case basis.

**Recommendation 9.1.** In very selected patients with OMD, TFI longer than 12–18 months and good performance status, a new liver resection may be considered according to the same oncological principles of recommendations 6.1–6.5.

Level of evidence: V

Strength of recommendation: B

Level of consensus: 100% (35) yes, 0% (0) no (35 voters)

**Recommendation 9.2.** Patients with liver recurrence should receive preoperative systemic treatment if they have not received adjuvant chemotherapy after the first resection, or on a case-by-case basis according to the principles of recommendation 11.2.

Level of evidence: V

Strength of recommendation: B

Level of consensus: 97% (34) yes, 3% (1) no (35 voters)

### 3.10. Which Patients Are Candidates for Ablative Techniques for Liver Metastases?

As previously discussed, surgical resection is associated with long-term survival benefit in carefully selected patients with liver-dominant metastatic disease [18,19,20]. In patients with limited metastases for whom surgery is not an option, multidisciplinary discussion should help identify candidates who may benefit from catheter-directed or local ablative therapies.

Regarding patient selection, the presence of unresectable liver-dominant metastases or medical comorbidities that preclude surgical resection may render patients candidates for local ablative therapies. Patients whose disease fails to respond to conventional therapies should also be considered for transarterial treatment [43]. For patients receiving systemic therapy for multiple metastases, ablation of liver-dominant lesions (i.e., the liver lesion with the highest risk of progression or rupture [44], a symptomatic lesion among multiple lesions [45,46], or a lesion that requires focal therapy despite the presence of other lesions [46]) after disease stability on chemotherapy can represent a well-tolerated maintenance therapy with significant progression-free survival (PFS) and systemic therapy-free intervals [43]. Absolute contraindications to focal therapies are few and include uncorrectable coagulopathy and active infection in the planned treatment area. Relative contraindications that need to be discussed considering the risk–benefit ratio are the inability to displace or protect adjacent critical structures and decompensated liver failure, considering also the specific treatment strategy) [43].

Several local therapy modalities are available, each requiring specific expertise.

Catheter-based therapies allow minimally invasive treatment of liver disease in selected patients, including those unable to tolerate surgical resection or with lesions not amenable to resection. Catheter-based intra-arterial therapies include transarterial embolization (TAE), transarterial chemoembolization (TACE), and transarterial radioembolization (TARE). These therapies have been shown to be safe for progressive disease, with high rates of local disease control and overall survival [47,48,49]. TACE has been shown to increase overall survival in sarcoma patients with liver metastases [18,50]. There is evidence to support the use of TARE with yttrium-90 beta particle-emitting microspheres in liver metastases from sarcoma [49]. There are also several options for tumor ablation, namely the application of thermal or non-thermal therapies to the tumor to achieve cell death. Thermal ablation can be achieved by hypothermia (cryoablation) or hyperthermia (radiofrequency ablation [RFA], microwave ablation, laser ablation, and high-intensity focused ultrasound—HIFU). Non-thermal ablation, such as irreversible electroporation (IRE), results in permanent cell membrane damage [51,52,53,54,55,56,57]. Local therapies such as cryoablation and RFA have been shown to improve overall survival in patients with oligometastatic disease [51,52,53,54,55,56,57]. There is strong evidence to recommend stereotactic body radiation therapy (SBRT) as a therapeutic tool with a favorable safety and toxicity profile for liver metastases from other malignancies. Several recent reviews (Table 2) and case series support the use of SBRT for local control, with potential survival benefits in selected patients with liver metastases from sarcoma [58,59,60,61].

**Recommendation 10.1.** Patients who are not candidates for surgery should be discussed in multidisciplinary meeting to evaluate their eligibility for nonsurgical local approaches.

Level of evidence: V

Strength of recommendation: B

Level of consensus: 97% (35) yes, 3% (1) no (36 voters)

**Recommendation 10.2.** Patients with limited liver metastases on systemic therapy should be considered for focal therapy of dominant liver lesions (i.e., any lesion that is either symptomatic or requires focal therapy) to achieve a systemic therapy-free interval or consolidate response to systemic therapy.

Level of evidence: V

Strength of recommendation: C

Level of consensus: 77% (27) yes, 23% (8) no (35 voters)

### 3.11. Is There a Role for Induction Chemotherapy Before Liver Metastasectomy?

The evidence for the use of induction or preoperative chemotherapy before liver metastasectomy in non-GIST sarcomas is very limited and based on retrospective studies (Table 3). In the aforementioned retrospective study by Goumard et al. [18], which evaluated 126 patients with non-GIST sarcoma with liver metastases who underwent liver surgery, 65 had received preoperative chemotherapy (median number of cycles: 6). Most patients had sarcoma of abdominal origin and 21% had extrahepatic disease at the time of liver metastasis diagnosis. Chemotherapy regimens included monotherapy with an anthracycline or an alkylating agent and combination therapy with an anthracycline plus an alkylating agent or docetaxel plus gemcitabine. Radiologic response to preoperative chemotherapy according to Choi (but not to RECIST 1.1) criteria was associated with a significant survival difference. The median OS for patients with partial response was 58.0 months versus 30.8 months for patients with disease progression (*p* = 0.04). Pathologic response was not significantly associated with survival, and the use of pre and/or postoperative chemotherapy did not impact OS or RFS. Other studies investigating the outcomes of liver metastasectomy in patients with STS did not provide detailed information about the preoperative chemotherapy regimens used or the specific outcomes associated with such treatments [17,19]. Older series of liver metastasectomy in several tumors, including sarcoma (mainly GIST), reported the use of chemotherapy but not the associated outcomes [41,63].

In the absence of robust evidence for preoperative chemotherapy prior to liver surgery, clinical trials and other retrospective studies addressing the role of chemotherapy in the neoadjuvant and metastatic settings can help predict treatment response and thus guide the therapeutic strategy. For example, regarding the choice between histotype-tailored neoadjuvant chemotherapy and standard non-histotype-tailored neoadjuvant chemotherapy, the ISG-STS1001 trial showed that there is no advantage of the latter in terms of DFS and OS, with the possible exception of high-grade myxoid liposarcoma (for which trabectedin showed equivalence to standard neoadjuvant chemotherapy in terms of clinical benefit) [64]. However, other endpoints, such as objective response rate (ORR), may be considered when discussing induction chemotherapy, as a higher rate of tumor shrinkage may be preferred to achieve better surgical outcomes. The choice of the chemotherapy regimen does not follow a formal indication, and the drugs used and the number of cycles are not defined.

Factors that should be considered when deciding on preoperative chemotherapy include the timing of metastasis (synchronous vs. oligorecurrent), intrinsic chemosensitivity of certain sarcoma subtypes, histologic grade, risk of local hepatic progression and loss of opportunity for R0 metastasectomy, symptomatic burden, and patient comorbidities. Regarding the number of cycles, an approach similar to that used in the neoadjuvant setting may be appropriate. The administration of at least three cycles appears to be noninferior to the administration of five cycles [65], and therefore, this panel suggests radiologic evaluation after two to three cycles to assess response.

The panel also believes that the timing of metastasis development and DFI in metachronous OMD are also important considerations when deciding whether to administer systemic treatment. Synchronous OMD is more likely to benefit from systemic therapy than oligorecurrent disease, as it is usually a more aggressive tumor and for a better understanding of the biological behavior of the newly diagnosed disease and a more accurate assessment of the risk-benefit ratio of radical surgery or ablative techniques.

Decisions regarding the use of perioperative systemic therapies should be discussed by a multidisciplinary tumor board with experience in both sarcoma treatment and liver surgery.

**Table 3 cancers-17-01295-t003:** Outcomes of preoperative chemotherapy before liver metastasectomy.

Reference	Sample Size (n)	Preoperative Treatment (n, %)	Agents of Preoperative Treatment	Outcomes
Goumard (2018) [18]	LMS: 59 Non-LMS: 67	LMS + non-LMS: 65 (52)	Anthracycline (n = 32), alkylating agent (n = 33), anthracycline plus alkylating agent (n = 26), gemcitabine plus docetaxel (n = 20), other (n = 6)	OS univariate analysis: HR 1.38 (0.86–2.23), *p* = 0.18 OS multivariate analysis: HR 1.57 (0.87–2.83), *p* = 0.13 RFS univariate analysis: HR 1.06 (0.71–1.57), *p* = 0.78
Tirotta (2020) [19]	24	3 (13%)	NR	NR
González-Abós (2023) [17]	Non-GIST: 12	4 (33%)	NR	NR
Pawlik (2006) [63]	LMS: 18; non-LMS (non-GIST): 12	14 (35%) *	Adriamycin-based chemotherapy	Sarcoma-only data not presented
Groeschl (2012) [41] *	Sarcoma: 98	55 (5%)	Sarcoma-only data not presented	NR

GIST, gastrointestinal stromal tumor; HR, hazard ratio; LMS, Leiomyosarcoma; N, number; NR, not reported; OS, overall survival; RFS, relapse-free survival. * Series included GIST.

**Recommendation 11.1.** Preoperative chemotherapy may be considered for patients with liver metastases amenable to surgical resection, after discussion in multidisciplinary tumor board, in centers with expertise in sarcoma treatment and liver surgery.

Level of evidence: V

Strength of recommendation: B

Level of consensus: 97% (35) yes, 3% (1) no (36 voters)

**Recommendation 11.2.** The decision regarding preoperative chemotherapy should be individualized, taking into account clinicopathologic factors including, but not limited to, intrinsic chemosensitivity, histologic grade, risk of local hepatic progression, symptom burden, and patient comorbidities.

Level of evidence: V

Strength of recommendation: B

Level of consensus: 97% (35) yes, 3% (1) no (36 voters)

### 3.12. Is There a Role for Adjuvant Systemic Treatment After Liver Metastasectomy?

As with preoperative chemotherapy, there is little evidence for the use of postoperative or adjuvant treatment after liver metastasectomy for STS. The few studies that mention the use of systemic adjuvant treatment are retrospective and often do not report the regimens used or the outcomes between groups (adjuvant chemotherapy vs. no chemotherapy).

Brudvik et al. [20] reported the use of postoperative chemotherapy in 27 (57%) cases of LMS and 28 (58%) cases of other non-GIST sarcomas. The regimens used included combinations of doxorubicin, docetaxel, gemcitabine, ifosfamide, and/or dacarbazine. In the LMS cohort, median RFS was 7.9 months in the adjuvant chemotherapy group versus 9.0 months in the no adjuvant chemotherapy group (*p* = 0.434; Table 4). In the non-GIST sarcoma cohort, median RFS was 7.4 months in the adjuvant chemotherapy group versus 9.4 months in the no adjuvant chemotherapy group (*p* = 0.805). Goumard et al. [18] also reported the use of postoperative systemic treatment but did not report the regimens used. In this series, the use of postoperative chemotherapy also did not appear to have an impact on RFS or OS in univariate analysis.

The role of systemic adjuvant treatment in the setting of localized STS has been addressed in several clinical trials (reviewed elsewhere [66]). This remains a challenging issue, particularly regarding patient (histologic) selection, identification of risk factors for recurrence and predictive factors of response, and choice of chemotherapy regimen. The development of static and dynamic nomograms for individualized prognostic prediction has contributed to this identification process [67]. However, these tools have not been validated in the (oligo)metastatic setting and cannot be recommended for treatment decisions in these patients.

The panel believes that the use of adjuvant chemotherapy in this setting cannot be formally recommended or discouraged, but should be discussed in multidisciplinary meeting, taking into account risk and prognostic factors as well as the predicted chemosensitivity of different histotypes.

**Table 4 cancers-17-01295-t004:** Outcomes of adjuvant (postoperative) treatment after liver metastasectomy.

Reference	Sample Size (N)	Adjuvant Treatment (N, %)	Agents of Postoperative Treatment	Outcomes
Brudvik (2015) [20]	LMS: 47 Non-LMS (non-GIST): 50	LMS: 27 (57) Non-LMS: 28 (58)	Combinations of doxorubicin, docetaxel, gemcitabine, ifosfamide, and/or dacarbazine	Adjuvant chemotherapy vs. no adjuvant chemotherapy Median RFS in LMS: 7.9 vs. 9.0 months, *p* = 0.434; in sarcoma: 7.4 months vs. 9.4 months, *p* = 0.805).
Goumard (2018) [18]	LMS: 59 Non-LMS: 67	LMS + non-LMS: 33 (26)	NR	OS univariate analysis: HR 1.27 (0.74–2.17), *p* = 0.39 RFS univariate analysis: HR 1.19 (0.76–1.86), *p* = 0.45
Grimme (2019) [21]	37	Lack of information on postoperative chemotherapy		
Tirotta (2020) [19]	24	12 (50%)	NR	NR
González-Abós (2023) [17] *	Non-GIST: 12	6 (50%)	Not specified (“Multiple lines of chemotherapy and monoclonal antibodies based on their chemosensitivity”.)	NR
Pawlik (2006) [63] *	LMS: 18; non-LMS (non-GIST): 12	3 (7.5%)	Adriamycin plus docetaxel, gemcitabine plus docetaxel, and cisplatin	Non-GIST sarcoma-only data not presented
DeMatteo (2001) [40] *	GIST/GI LMS: 34 (61%) Other sarcoma: 22 (39%)	23 (41%)	Not specified (“Patients received adjuvant chemotherapy at some point in their therapy”.)	NR
Adam (2006) [68] *	Sarcoma *: 125	NR	Sarcoma-only data not presented	NR
Groeschl (2012) [41] *	Sarcoma: 98	54 (56%)	Sarcoma-only data not presented	NR

GI, gastrointestinal; GIST, gastrointestinal stromal tumor; HR, hazard ratio; LMS, leiomyosarcoma; N, number; NR, not reported; OS, overall survival; RFS, relapse-free survival. * Series included GIST.

**Recommendations 12.1.** The recommendation of adjuvant systemic treatment for patients with STS with liver metastases whose primary neoplasm and liver metastases have been resected should be discussed in multidisciplinary meeting, taking into account existing risk factors.

Level of evidence: V

Strength of recommendation: B

Level of consensus: 100% (35) yes, 0% (0) no (35 voters)

### 3.13. How Should Follow-Up Be Performed After Metastasectomy?

Considering the available data, it is extremely difficult to clearly define the prognosis of STS patients after liver metastasis resection. In this context, two important facts should be highlighted. First, OS has improved significantly in recent years due to improvements in surgical techniques, patient selection, and the use of combined therapies. In their analysis of data from the Netherlands Cancer Registry of synchronous metastases of sarcomatous origin, Vos et al. showed an improvement in median OS from 5.8 to 8.1 months from 1989–1994 to 2010–2014 [69]. These data are consistent with the results of the Transatlantic Australasian Retroperitoneal Sarcoma Working Group (TARPSWG) analysis of liver metastasectomy from 1997 to 2015 and the data shown in Table 1. Second, it seems clear that although it is difficult to establish a fixed 5-year OS rate for liver resection for STS metastases to the liver, the data obtained in different series are always better than any other therapy applied.

In a recent review by the TARPSWG, despite the small sample size (12 series from 1997 to 2015 with a total of 497 patients and an average of 45 patients per series), 5-year OS ranged from 20% to 49% [70]. These data are consistent with the studies shown in Table 1, with median 5-year OS ranging from 18% to 53.1%. Regarding PFS and DFI, the median values ranged from 6.9 to 33 months in the considered series. Of note, while PFS is usually used as a clinical endpoint in advanced disease, DFI is usually used in the setting of definitive treatment (either surgical or non-surgical); in this case, these two endpoints were considered interchangeably, as this is a setting of advanced disease treated with radical intent.

The review of the available literature revealed a lack of clinical guidelines with specific and clear recommendations regarding follow-up in the setting of liver metastasis resection. Several single-institution studies have described their follow-up protocols, but without mentioning their rationale. Considering OS and PFS/DFI data, the panel proposes a follow-up schedule based on physical examination and ultrasound, and oral and intravenous contrast-enhanced computed tomography (CT) of the chest and abdomen every 3 to 4 months for the first 2 years and every 6 months thereafter until the fifth year. Other radiological investigations should be considered on a case-by-case basis, taking into account the histotype, symptoms, and clinical course of the disease.

**Recommendation 13.1.** Patients should be evaluated by physical examination, abdominal ultrasound, and (oral and intravenous) contrast-enhanced computed tomography of the chest and abdomen every 3 to 4 months for the first 2 postoperative years, every 6 months through the end of the fifth year, and annually thereafter.

Level of evidence: V

Strength of recommendation: B

Level of consensus: 86% (31) yes, 14% (5) no (36 voters)

**Recommendation 13.2.** Further radiologic evaluation should be considered on a case-by-case basis depending on histotype, pattern of recurrence, and clinical course.

Level of evidence: V

Strength of recommendation: B

Level of consensus: 100% (36) yes, 0% (0) no (36 voters)

## 4. Clinical Consensus Recommendations for the Management of Liver Metastases from Soft Tissue Sarcoma

The 34 recommendations proposed by the panel that were voted by sarcoma experts in the I Ibero-American Consensus Conference on the Management of Liver Metastases from Soft Tissue Sarcomas are presented in Table 5. Consensus was reached (>80% agreement) on more than 90% of the proposed recommendations. Three (8.8%) of the proposed recommendations did not reach consensus, seven (20.6%) reached a weak consensus, 12 (35.9%) a strong consensus and 12 (35.9%) an unanimous consensus.

## 5. Discussion

This review summarized the available evidence regarding the treatment of liver metastases from non-GIST STS. Given the lack of specific data on the topic, a definition of OMD in the context of STS is proposed relying on broader cancer definitions. Although the authors sought to define a number of lesions and affected organs and to exclude diffuse organ involvement, the concept of OMD will likely continue to evolve as the understanding of sarcoma biology, genomics, metabolic pathways and tumor microenvironment and its interaction with the immune system advances.

The timing of OMD diagnosis (either synchronous or metachronous) also emerged as an important issue, as it appears to be a prognostic factor that could influence both therapeutic decisions and the need for tumor biopsy. This factor may reflect tumor behavior and biology and, although a simplistic marker, remains one of the most accessible clinical parameters when selecting the most appropriate treatment strategy. As of today, the timing of recurrence continues to play a critical role in determining whether the disease should be managed with radical or palliative intent.

Regarding the role of biopsy, although it has been proposed to perform liver metastasis biopsy when systemic treatment is the first-line option, no consensus has been reached. This may be due to the low probability of obtaining an alternative diagnosis, together with the considerable risk associated with the procedure in several cases, which discourages the pursuit of histologic confirmation. In the revised series, the need for biopsy was often not reported or not required, reflecting the lack of data in this setting.

Radiologic evaluation (the role of CT and MRI) and preoperative patient assessment (FLR, comorbidities, performance status, ASA physical status classification system) were also debated for the selection of best candidates for liver resection. Data are mostly extrapolated from other contexts of HPB surgery. Regarding the indications for liver surgery, consensus was reached on all recommendations except one. The proposal to discourage surgical liver resection in the setting of bilobar liver metastases and when surgery would require complex hepatectomy (e.g., vascular involvement, central location) was not widely accepted. This may be due to the interpretation of bilobar metastasectomy as a complex procedure, when in fact what was meant was the setting of bilobar disease and the need for complex hepatectomy due to other factors such as vascular involvement or central location. The decision to perform a more complex and risky surgery may also depend on the expertise of the surgical team and the experience of the center. On the other hand, there was a strong consensus on the best surgical approach: R0 resection by an experienced HPB center, considering anatomic and oncological principles to define the extent of surgery, allowing major hepatectomy in selected cases and considering MIS when technically feasible.

The role of systemic treatment in the (neo)adjuvant setting of radical treatment remains poorly defined. However, consensus was reached that each patient should be evaluated individually by a multidisciplinary team of sarcoma experts, considering clinicopathological factors. Remetastasectomy may also be considered, considering prior abdominal surgeries and DFI. Although no specific studies have accessed the benefit of subsequent liver metastasectomy, some series have reported the number of patients who underwent previous liver resection. While specific outcomes for this subset of patients were not reported, the authors agree that remetastasectomy should be considered and discussed as a potential treatment option.

Regarding local therapies, there is an increasing diversity of tumor ablation techniques. In general, the results reported reflect single-institution experiences, based on the available resources and expertise at each center. Proficiency in each technique is essential to achieve optimal outcomes.

Sarcoma centers are encouraged to develop and implement local therapies best suited to their patient cohort, as an expanding body of evidence supports their benefit in oligometastatic disease (OMD) patients.

Furthermore, close follow-up is crucial to promptly identify local recurrence or new oligometastases that may be amenable to radical treatment. Follow-up also plays a key role in monitoring adverse events and treatment sequelae. The present recommendations were based on hepatic surgery principles, considering the DFI reported in the reviewed studies and the overall prognosis of mSTS. However, due to the lack of clear, evidence-based guidelines on this topic, further research is warranted.

As a rare metastatic site of an already rare disease, evidence regarding STS liver metastases is scarce and primarily based on case series and small retrospective cohorts. The available data are highly heterogeneous and often inconclusive, making it challenging to draw robust clinical conclusions. This heterogeneity can be attributed to several factors: (i) limited sample size—most published series included a small number of patients; (ii) wide time frames—many studies span decades, often including patients diagnosed between the 1990s and 2019, making treatment comparisons difficult; (iii) lack of data specificity—in some studies, it is difficult, if not impossible, to isolate data specifically related to liver metastases of sarcomatous origin, as some patients are broadly categorized as non-GIST; (iv) histologic variability—some studies focused exclusively on a single histologic subtype, such as LMS, precluding comparative analyses; (v) lack of randomized trials and prospective studies. Notably, LMS has a better prognosis than other STS histologies, partly due to its greater chemosensitivity, which may bias cohort outcomes; (v) inclusion of GIST tumors—some series included GISTs, which have a fundamentally different prognosis and treatment; and (vi) diverse treatment approaches—patients included in different studies received highly variable treatments, including chemotherapy, RT, local percutaneous ablative therapies, or multimodal combinations, adding another layer of complexity to data interpretation. In the absence of prospective studies and randomized data, these recommendations are based primarily on small retrospective cohorts, and principles regarding surgical approach, extent of hepatectomy and metastasectomy, and patient assessment have been extrapolated from more common malignancies such as CRC and neuroendocrine tumors.

Patient selection is key, and the development of biomarkers that could help to predict which patients would most likely benefit from focal or systemic therapy is crucial. For example, the identification of patients that could benefit from adjuvant systemic therapy is likely to be refined, in short time, by the widespread use of circulating tumor DNA (ctDNA) levels monitorization in clinical practice. Consistent data suggest that ctDNA may be used as an extremely sensitive instrument to spot patients with significant risk of recurrence following curative treatments and that may be better candidates for adjuvant systemic treatment [71,72,73,74,75,76,77]. ctDNA levels at diagnosis seem to be strongly correlated with known clinical risk factors and negative outcomes, while its levels variation throughout the treatment timeline are correlated with treatment response [71,72,73,74,75,76,77]. Indeed, ctDNA detection and its levels variation following definitive treatment have a strong potential of allowing real-time monitoring of minimal residual disease activity, permitting pre-clinically apparent relapse detection and guiding an early systemic adjuvant treatment start [72,73,74,75,76,77]. However, evidence is still short to allow formal recommendations on the use of ctDNA levels monitorization as a tool to guide decisions on adjuvant systemic treatment. Although, these studies address the setting of adjuvant therapy in localized disease, the promising results bring hope that ctDNA may have a role in patient selection and in guiding treatment strategy in OMD in a future to come.

Regarding novel therapies in STS, namely immunotherapy, most clinical trials for the use immune-check-point inhibitors in STS included a diverse range of STS histotypes and the results have been overall disappointing, with response rates around 15% [78]. However, significant variations in tumor response rates have been observed across different histologies. These findings may indicate that for some histologies, immunotherapy could became a standard of care in the near future. The use of ICIs in metastatic sarcoma, could provide new settings where initial (partial/complete) remission is complemented with focal therapy for oligopersistent or oligorecurrent disease, while the patient maintains an effective and well-tolerated treatment.

Despite addressing a highly specific subset of a rare disease, for which limited data is available in the literature, the panel was able to reach consensus (>80% agreement) on more than 90% of the proposed recommendations. This high level of agreement likely reflects the panel’s composition of dedicated sarcoma specialists who share an up-to-date and expert-driven perspective on the topic. Of note, the meeting included sarcoma and HPB surgery experts from Portugal, Spain, and several Latin American countries, representing a diverse range of centers and clinical backgrounds.

## 6. Conclusions

Overall, the authors believe that consensus guidelines play a critical role in addressing topics with limited robust evidence, providing valuable guidance to clinicians in the management of challenging, out-of-protocol cases. However, there is an urgent need for more robust data and prospective studies. Collaborative research efforts between centers and integration into international research networks could be highly beneficial in advancing knowledge and improving patient care.

## Figures and Tables

**Table 1 cancers-17-01295-t001:** Studies reporting outcomes of patients with resected liver metastases from soft tissue sarcoma.

Reference	Sample Size (N) and Study Period	Median DFI or PFS (Months)	Overall Survival at 1 Year	Overall Survival at 5 Years	Median Overall Survival (Months)
Brudvik (2015) [20]	LMS: 47 Non-LMS (non-GIST): 50 1998–2013	LMS: 7.9 Non-LMS: 8.8	NR	LMS: 48.4% Non-LMS: 44.9%	NR
Goumard (2018) [18]	LMS: 59 Non-LMS: 67 1998–2015	12 (0–298)	98% 83%	52% 47%	58
Grimme (2019) [21]	37 1998–2014	16 (1–61)	86%	42%	46 (1–161)
Tirotta (2020) [19]	24 2009–2019	33 (0–208)	94%	18%	35 (10–113)
Outani (2022) [34]	89 2000–2018	NA	2-y OS: 51.1%	24.4%	25 (13–34)
González-Abós (2023) [17]	Non-GIST: 12 2003–209	27 (5−27)	100%	42.9%	38
Burkhard-Meier (2024) [35] *	40 * 2017–2021	6.9 (5.9–8.5) **	93.7% **	53.1% **	64.8 (52.8–93.6) **

* Includes surgery and non-surgical local ablative techniques; only 16% of treated lesions were liver metastases. ** Data regarding the entire cohort studied (different metastatic sites). DFI, disease-free interval; GIST, gastrointestinal stromal tumor; LMS, leiomyosarcoma; N, number; NA, not accessible (abstract only); NR, not reported; OS, overall survival; PFS, progression-free survival; y, years.

**Table 2 cancers-17-01295-t002:** Clinical outcomes of local therapies for liver metastases.

Reference	Local Therapy	Sample Size (N) and Study Period	Progression-Free Interval (Months)	Overall Survival at 1 Year	Overall Survival at 2 Years	Overall Survival (Months)
Maluccio (2006) [47]	TAE	24 1996–2002	22	61%	41%	NR
Chapiro (2015) [48]	TACE	50 2000–2013	6,3	NR	NR	21.2
Miller (2018) [49]	TARE	39 2006–2015	NR	NR	NR	30.0
Awad (2024) [62]	RFA	55 2011–2021	NR	66%	40%	NR
Bourguoin (2022) [52]	MWA/Cryoablation	27 2009–2021	NR	100%	89%	NR
Rodríguez (2024) [61]	SBRT	52 2015–2018	82.9% at 2 y 76.5% at 3 y	NR	NR	27.7

MWA, microwave ablation; N, number; NR, not reported; RFA, radiofrequency ablation; SBRT, stereotactic body radiation therapy; TACE, transarterial chemoembolization; TAE, transarterial embolization; TARE, transarterial radioembolization; y, years.

**Table 5 cancers-17-01295-t005:** Summary of recommendations for the management of liver metastases from soft tissue sarcoma.

Recommendations and Levels of Evidence	Total (n)	Yes (n)	No (n)	Consensus
**1. What is the definition of oligometastatic disease in soft tissue sarcoma?**				
**1.1.** OMD in STS should be defined by mSTS with a number of metastatic lesions ≤ 5 in ≤ 2 different metastatic sites, confirmed by morphologic staging and additional imaging and/or nuclear medicine modalities, as appropriate, in which the primary and all metastatic sites are amenable to radical treatment (either surgical and/or nonsurgical ablative procedures) (V,B).	36	33	3	92% strong
**1.2.** Involvement of a diffuse organ such as malignant pleural, pericardial, peritoneal, or leptomeningeal sarcomatosis should exclude patients from being considered as oligometastatic, as this scenario would render the disease ineligible for radical treatment (V,B).	34	30	4	88% weak
**2. What is the definition of synchronous and metachronous OMD?**				
**2.1.** Synchronous OMD is defined as OMD detected either at the initial diagnosis of the primary tumor or within six months thereafter, with a limited number of metastases detected simultaneously, with a disease burden of ≤5 metastatic lesions in ≤2 different metastatic sites (V,B).	36	30	6	83% weak
**2.2.** Metachronous OMD is defined as a OMD detected more than 6 months after initial diagnosis, with a limited number of metastases detected simultaneously, with a disease burden of ≤5 metastatic lesions in ≤2 different metastatic sites (V,B).	37	30	7	81% weak
**3. What is the role of biopsy in sarcoma liver metastases?**				
**3.1.** In resectable metachronous liver metastases, biopsy should not be mandatory but could be considered if the tumor origin is uncertain (e.g., long DFI, risk factors for hepatocellular carcinoma, metachronous colorectal cancer) or if a preoperative systemic treatment approach is proposed (V,B).	37	36	1	97% strong
**3.2.** If synchronous OMD and induction systemic treatment is planned, biopsy should be performed prior to treatment initiation (V,B).	37	23	14	62% not reached
**3.3.** A minimum of four large-gauge cores (14–16 G) is recommended to ensure adequate tissue sampling (IV,A).	36	32	4	89% weak
**4. How should sarcoma liver metastases be evaluated radiologically?**				
**4.1.** Iodinated contrast-enhanced CT scan is recommended to evaluate vascular involvement and the abdominal cavity (especially in RPS) (V,B).	36	36	0	100% unanimous
**4.2.** Hepatic MRI is recommended for diagnosis and evaluation of lesions smaller than 1 cm and for characterization of lesions of uncertain origin (V,B).	36	36	0	100% unanimous
**4.3.** ^18^F-FDG-PET/CT is recommended to exclude extrahepatic metastases in appropriate subtypes (e.g., lymph node metastases) (V,B).	38	32	6	84% weak
**5. In which patients with mSTS is hepatic metastasectomy indicated?**				
**5.1.** Surgical resection of liver metastases should always be previously discussed within a multidisciplinary tumor board at a sarcoma referral or network center. It is important to weigh the risks of surgery against the expected benefits (V,B).	38	38	0	100% unanimous
**5.2.** Surgical resection of liver metastases should only be performed in reference or specialized high-volume centers with experience in liver surgery (V,B).	35	34	1	97% strong
**5.3.** For metachronous disease, liver resection should be considered in patients diagnosed with STS and oligometastatic disease confined to the liver and amenable to complete surgical resection (V,B).	36	36	0	100% unanimous
**5.4.** In highly selected patients with metachronous OMD with a favorable prognostic profile that includes liver metastases and limited extrahepatic disease (lung only), surgical resection may also be considered if all lesions are potentially resectable (V,C).	37	34	3	92% strong
**5.5.** Surgical resection of the liver for OMD of STS should be discouraged if the patient presents with bilobar liver metastases requiring complex hepatectomies (e.g.,with vascular involvement, or central location) (V,D).	35	20	15	57% not reached
**6. What is the best surgical approach?**				
**6.1.** The primary goal of resection of sarcoma liver metastases should be to achieve an R0 resection (V,B).	36	35	1	97% strong
**6.2.** The feasibility of liver surgery to achieve an R0 resection should be evaluated by an experienced HPB surgeon in a reference center for HPB pathology or surgical oncology (V,B).	36	35	1	97% strong
**6.3.** The extent of surgery should be determined by anatomic factors (location of lesions) and oncological principles (R0 resection), favoring parenchyma-sparing resection (V,B).	36	36	0	100% unanimous
**6.4.** Major hepatectomy may be justified in selected cases with good prognostic features, with individualization of the approach based on the specific characteristics of each case (V,B).	36	34	2	94% strong
**6.5.** Minimally invasive liver resections (laparoscopic or robotic) may be considered when technically feasible, always with the goal of achieving an R0 resection (V,B).	35	35	0	100% unanimous
**7. How should patients be evaluated before surgery?**				
**7.1.** When planning a major hepatectomy, preoperative assessment of FLR should follow recommendations derived from experience in other contexts where major hepatectomy is performed (V,B).	37	37	0	100% unanimous
**7.2.** Preoperative patient assessment should follow the principles of any major surgery (assessment of comorbidities, performance status, and ASA physical status) (V,B).	37	37	0	100% unanimous
**8. When should surgical resection of liver metastases be considered?**				
**8.1.** In patients with STS OMD, surgery for liver metastases could be considered for metachronous lesions (DFI ≥ 6 months) as upfront treatment or after systemic therapy (especially in subtypes with better response to chemotherapy) (V,B).	36	34	2	94% strong
**8.2.** In patients with STS OMD and synchronous disease, systemic treatment (chemotherapy, etc.) should be the preferred front-line approach (V,B).	35	30	5	86% weak
**8.3.** In patients with STS OMD and synchronous disease, liver surgery may be considered for hepatic residual disease if there is a good response to systemic treatment (at least stable disease) and the primary tumor is resectable (V,B).	34	34	0	100% unanimous
**9. Remetastasectomy: Is there a role for surgery for liver recurrence?**				
**9.1.** In very selected patients with OMD, TFI longer than 12–18 months, and good performance status, a new liver resection may be considered according to the same oncological principles of recommendations 6.1–6.5 (V,B).	35	35	0	100% unanimous
**9.2.** Patients with liver recurrence should receive preoperative systemic treatment if they have not received adjuvant chemotherapy after the first resection, or on a case-by-case basis according to the principles of recommendation 11.2 (V,B).	35	34	1	97% strong
**10. Which patients are candidates for ablative techniques for liver metastases?**				
**10.1.** Patients who are not candidates for surgery should be discussed in multidisciplinary meeting to evaluate their eligibility for nonsurgical local approaches (V,B).	36	35	1	97% strong
**10.2.** Patients with limited liver metastases on systemic therapy should be considered for focal therapy of dominant liver lesions (i.e., any lesion that is either symptomatic or requires focal therapy) to achieve a systemic therapy-free interval or consolidate response to systemic therapy (V,C).	35	27	8	77% not reached
**11. Is there a role for induction chemotherapy before liver metastasectomy?**				
**11.1.** Preoperative chemotherapy may be considered for patients with liver metastases amenable to surgical resection, after discussion in multidisciplinary tumor board, in centers with expertise in sarcoma treatment and liver surgery (V,B).	36	35	1	97% strong
**11.2.** The decision regarding preoperative chemotherapy should be based on a personalized approach and take into account clinicopathologic factors including, but not limited to, intrinsic chemosensitivity, histologic grade, risk of local hepatic progression, symptom burden, and patient comorbidities (V,B).	36	35	1	97% strong
**12. Is there a role for adjuvant systemic treatment after liver metastasectomy?**				
**12.1.** The recommendation of adjuvant systemic treatment for patients with STS with liver metastases whose primary neoplasm and liver metastases have been resected should be discussed in multidisciplinary meeting, taking into account existing risk factors (V,B).	35	35	0	100% unanimous
**13. How should post-metastasectomy follow-up be performed?**				
**13.1.** Patients should be evaluated by physical examination, abdominal ultrasound, and (oral and IV) contrast-enhanced computed tomography of the chest and abdomen every 3 to 4 months for the first 2 postoperative years, every 6 months until the end of the fifth year, and annually thereafter (V,B).	36	31	5	86% weak
**13.2.** Further radiologic evaluation should be considered on a case-by-case basis depending on histotype, recurrence pattern, and clinical course (V,B).	36	36	0	100% unanimous

ASA, American Society of Anesthesiologists; CRC, colorectal cancer; DFI, disease-free interval; ^18^F-FDG-PET/CT, positron emission tomography with 2-deoxy-2-[fluorine-18]fluoro-D-glucose integrated with computed tomography; FLR, future liver remnant; HPB, hepato-pancreato-biliary; MRI, magnetic resonance imaging; mSTS, metastatic soft tissue sarcoma; OMD, oligometastatic disease; RPS, retroperitoneal sarcoma; STS, soft tissue sarcoma; TFI, treatment-free interval.

## Data Availability

The original contributions presented in this study are included in the article.

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
