# Peer review of "Ibero-American Consensus for the Management of Liver Metastases of Soft Tissue Sarcoma: Updated Review and Clinical Recommendations"

_cancers, 2025, doi:10.3390/cancers17081295_

Round 1
Reviewer 1 Report
Comments and Suggestions for Authors
Very well written. Despite the incidence of the pathology, it remains one of the most challenging ones.
Although very good, an underlined critical issue for managing a patient with metastatic disease, seems to be only a surgical approach. For an oncologist, probably also some suggestions regarding the standard chemo regimens could be helpful. Also think of the precise maximum limit of chemo administration before any local intervention, to avoid loss of resectability for borderline resectable cases.
Author Response
Reviewer 1
“Very well written. Despite the incidence of the pathology, it remains one of the most challenging
ones.”
Author response: Thank you, it is indeed a very challenging topic in the treatment of soft tissue sarcoma.
- Comment from Reviewer 1: “Although very good, an underlined critical issue for managing a patient with metastatic disease, seems to be only a surgical approach. For an oncologist, probably also some suggestions regarding the standard chemo regimens could be helpful. Also think of the precise maximum limit of chemo administration before any local intervention, to avoid loss of resectability for borderline resectable cases.
Author response: Thank you for your input. In the setting of induction chemotherapy, we have revised the text to better explain the factors weighing in in the decision of the chemotherapy regimen and the appropriate number of cycles. About the risk of losing resectability in borderline resectable cases, factors such as chemosensitivity and timing of appearance of metastases will have a major contribution to the discussion, but in the end, it will be a personalized decision that should include the patient, after explaining the expected risks and benefits. Regarding adjuvant chemotherapy after liver metastasectomy, we have addressed the lack of evidence in this setting in “Section 3.12 Is there a role for adjuvant systemic treatment after liver metastasectomy?” and mentioned the most commonly used cytotoxic drugs, that are, in fact similar to the regimens recommended in the adjuvant setting for localized disease. Lastly, we have also addressed the use of nomograms in this scenario, that we could not recommend as the available nomograms were validated only for localized disease. We have revised the manuscript accordingly.
Reviewer 2 Report
Comments and Suggestions for Authors
Please refer attachment.

Author Response
Reviewer 2
“Multidisciplinary expert group reviewed the literature and established clinical recommendations for managing liver-only STS metastases, emphasizing surgical and ablative techniques as potential long-term disease control options. The recommendations, developed through an Ibero-American consensus meeting, focus on patient selection, surgical strategies, and the integration of systemic and local treatments to improve survival outcomes.”
- Comment (1) from Reviewer 2: “Introduction: The introduction lacks a structured discussion on why liver metastases in STS present a unique challenge. A more explicit discussion on current treatment gaps and why a consensus was needed would add value. It should explain the rationale behind conducting the Ibero-American Consensus and how it contributes to existing literature
Author response: Thank you for your comment. We have made changes to the Introduction to better explain why this is a pertinent and challenging topic and why it merits a consensus meeting to address treatment gaps.
- Comment (2) from Reviewer 2: “Methods: More details are needed on how the expert panel was selected, the voting methodology, and how disagreements were resolved. A clearer description of how the literature review was conducted, including databases searched and inclusion/exclusion criteria, would improve reproducibility.”
Author response: We have added more details to the Methods section to addressed all these concerns and help to improve reproducibility.
- Comment (3) from Reviewer 2: “Definition of Oligometastatic Disease: The definition of OMD varies across oncology literature. The manuscript should explicitly define it and justify the chosen criteria (≤5 lesions in ≤2 sites). Consider adding a comparison with other existing definitions from
oncology guidelines (e.g., EORTC, ESTRO).
Author response: Thank you for your comment. Regarding the topic of OMD, specifically in the setting of soft tissue sarcomas, no specific definition exists in oncology guidelines, besides what is mentioned in the text (“intermediate state where local or treated metastases control may improve systemic disease control, including the potential to delay (or avoid) the systemic spreading of the disease and the use of systemic chemotherapy”)
As suggested, we added a justification for the chosen criteria: “As no definition exists for STS OMD, we propose the definition of five or less lesions and two or less affected organs, considering that the most common site of metastization is the lung (even for intra-abdominal sarcomas) and that in patients with liver metastases, the presence of concurrent extra-pulmonary extra-hepatic sites could mean a more systemic disease and portend a dismal prognosis, making it less likely that the patient could be benefit from focal therapy.”
- Comment (4) from Reviewer 2: “Role of Biopsy in Liver Metastases: The manuscript presents recommendations but lacks discussion on biopsy techniques, accuracy, and risks. Should clarify scenarios where biopsy is mandatory vs when it can be avoided.”
Author response: Thank you for your feedback. The role of biopsy in soft tissue sarcoma is a crucial topic. However, there is a lack of evidence and recommendations regarding the role, timing, techniques, accuracy and risks of biopsy in the setting of metastatic disease amenable to local treatment. We reviewed this subject and have added additional information to the manuscript (Section 3.3). Regarding the scenarios where liver biopsy should be considered/mandatory or could be spared, we believe that we have addressed this matter based on the available literature, in our manuscript.
- Comment (5) from Reviewer 2: “Imaging Recommendations: The manuscript correctly recommends MRI and PET-CT but should elaborate on the strengths and limitations of each modality. Consider including a comparative table for imaging techniques with sensitivity/specificity data for STS liver metastases.”
Author response: We sincerely appreciate your suggestion to elaborate on the strengths and limitations of each imaging modality. To address this matter, we have included a paragraph summarizing the advantages, limitations, and diagnostic performance (sensitivity/specificity) of CT, MRI, and 18F-FDG PET/CT for detecting liver and extra liver metastases in STS. Thank you for your thoughtful suggestion.
- Comment (6) from Reviewer 2: “Role of Induction Chemotherapy and Adjuvant Treatment: The manuscript lacks strong evidence for preoperative and postoperative chemotherapy but still makes recommendations. It should clarify when systemic therapy is strongly advised vs when it
remains optional. Data on histotype-specific chemosensitivity should be added (e.g., leiomyosarcoma vs liposarcoma).”
Author response: Thank you for your input. Indeed, there is a lack of robust data regarding the use of systemic therapy either before or after radical treatment of metastases. We have revised the section on Induction Chemotherapy to make it clearer that no formal indication exists and that the evaluation of ChT being strongly advised or optional should be made on a case-by-case basis, taking into account several factors (described in the text). We also emphasized the timing of metastization and the DFI in OMD as strong factors to weigh in (synchronous/shorter DFI in favor of preoperative systemic therapy). Regarding adjuvant treatment, we presented the available data and proposed a recommendation that adjuvant treatment should also be discussed on a case-by-case basis, integrating risk factors for recurrence. We have revised the text in this section to include the histotype chemosensitivity as a factor to be considered in the decision.
- Comment (7) from Reviewer 2: “Limitations and Future Research: The manuscript should acknowledge the lack of randomized trials and prospective data. Areas requiring further research (e.g., biomarkers for patient selection, role of immunotherapy) should be discussed.”
Author response: Thank you for your suggestions. Regarding limitations, we address the lack of robust evidence – please see 5. Discussion and Concluions “As a rare metastatic site, (…) extrapolated from more common malignancies such as colorectal cancerCRC and neuroendocrine tumors.” However, as suggested, we specifically added the lack of randomized trials and prospective data in this section.
We added data on ctDNA as possible future biomarker for patient selection in the adjuvant setting and discussed its possible role in oligometastatic disease (OMD). We also mentioned the role of immunotherapy in sarcomas and how it could change the landscape of OMD treatment.
- Minor Comment (1) from Reviewer 2: “"et al." Rectify the same in entire manuscript.”
Author response: We agree with the reviewer’s comment. Accordingly, we have reviewed and corrected this expression throughout the manuscript.
- Minor Comment (2) from Reviewer 2: “Line 255: "metastases". Rectify the spelling in manuscript.”
Author response: We revised the manuscript to make sure metastasis (singular) and metastases (plural) are adequately used.
- Minor Comment (3) from Reviewer 2 asking to change “OMD is considered as an intermediate state in which local or treated metastases control may yield improved systemic control." to "OMD is considered an intermediate state where local treatment of metastases may improve systemic disease control.".
Author response: We addressed this sentence and changed it accordingly.
Reviewer 3 Report
Comments and Suggestions for Authors
The text discusses liver metastasis of sarcoma.
While very interesting, there are several concerns that need to be discussed.
Liver metastases are very rare, but when they are detected, they are terminal,
What do you think?
Liver metastases are unlikely to be detected unless imaging studies are performed.
However, they could potentially be present. What is your opinion on this?
What kind of patients are most likely to develop liver metastases?
Are they related to inflammation?
Favorable.
Author Response
Reviewer 3
The English could be improved to more clearly express the research.
Author response: We had the manuscript revised by a medical writer.
“The text discusses liver metastasis of sarcoma. While very interesting, there are several concerns that need to be discussed.”
- Comment (1) from Reviewer 3: “Liver metastases are very rare, but when they are detected, they are terminal, What do you think?”
Author response: Thank you for your question. As we mentioned in the Introduction, the prognosis of liver metastases in non-GIST soft tissue sarcoma (STS) is poor (e.g. the 2-year survival rate after the diagnosis of liver metastatic disease is around 20% [Ref. 5]; in another series [Ref. 7], the reported median survival time after the diagnosis of hepatic metastases was 12 months). Despite this dim prognosis, there is enough evidence to support surgery in these patients as systemic therapy (e.g. chemotherapy) rarely (if ever) leads to complete or sustained remission and, reported series of liver metastasectomy for STS show a benefit in survival outcomes.
- Comment (2) from Reviewer 3: “Liver metastases are unlikely to be detected unless imaging studies are performed. However, they could potentially be present. What is your opinion on this? What kind of patients are most likely to develop liver metastases? Are they related to inflammation?”
Author response: Thank you for pointing this out. In “Section 3.1 What is the definition of oligometastatic disease for sarcoma?” we discuss the definition of oligometastatic disease in sarcoma patients, as these are the potential candidates for surgical or ablative treatment in the metastatic setting. As explained in this section, patients should be adequately staged to rule out widespread disease and confirm the oligometastatic stage. Furthermore, in “Section 3.4. How should sarcoma liver metastases be radiologically evaluated?”, we discuss the role of imaging in staging and the appropriate modalities to address specific concerns, specifically the role of 18F-FDG PET in excluding extra-hepatic disease. Considering that the patients at higher risk of developing liver metastases have a diagnosis of retroperitoneal or intra-abdominal sarcoma, special attention should be given to liver staging in these patients.
Regarding the development of liver cancer, we know that non-resolving inflammation plays a central role in the tumorigenesis of hepatocellular carcinoma, as the vast majority of these arise in a background of chronic liver disease and cirrhosis. In secondary hepatic lesions, the role of pro-inflammatory signaling is also thought to be present both as a crucial step in eliminating invading cancer cells but, on the other hand, also helping to promote a pro-metastatic environment, metastatic seeding and colonization [2]. However, these mechanisms of tumorigenesis and the interaction between inflammation and metastatic spread are better characterized for colorectal cancer, due to its prevalence and tropism for liver spreading. In STS, the mechanisms behind liver metastization are not so well defined, being a rare and heterogeneous group of diseases and much rarer presentation with liver secondary lesions.
References:
[1] Yu, L.-X.; Ling, Y.; Wang, H.-Y. Role of nonresolving inflammation in hepatocellular carcinoma development and progression. npj Precision Oncology 2018, 2, 6, doi:10.1038/s41698-018-0048-z.
[2] Strathearn, L.S.; Stepanov, A.I.; Font-Burgada, J. Inflammation in Primary and Metastatic Liver Tumorigenesis–Under the Influence of Alcohol and High-Fat Diets. Nutrients 2020, 12, 933.
Reviewer 4 Report
Comments and Suggestions for Authors
The methodology is appropriate, and the review is based on up-to-date literature. The language is clear, and the article is well-structured and easy to follow.
However, I have a few minor remarks:
- In section 3.4, point 4.2, the authors recommend, “Hepatic MRI is recommended to diagnose and evaluate lesions smaller than 1 cm and to characterize lesions of uncertain origin.” While this is valid, MRI has been shown to be superior to CT in detecting additional liver lesions, beyond those initially identified by CT. This is particularly relevant for patients being considered for liver resection. Therefore, I believe that MRI should be mandatory in the preoperative workup for all patients undergoing liver metastasectomy, not just for those with lesions smaller than 1 cm.
- In Recommendation 4.3, the authors state, “18F-FDG-PET/CT scan should be recommended to rule out extrahepatic metastases in suitable subtypes (e.g., lymph node metastases).” I believe PET/CT should be recommended for all patients considered for liver metastasectomy, not just for specific subtypes. This would help exclude other metastatic sites and detect potential recurrences, thereby optimizing patient selection for surgery.
- In Recommendation 5.5, the authors propose, “Liver surgical resection of OMD of STS should be discouraged if the patient presents with bilobar liver metastatic disease.” However, I do not believe bilobar disease should be considered an absolute contraindication. For instance, in cases where two bilobar metastases can be managed with parenchymal-sparing surgery, resection remains a viable option. Additionally, a multimodal approach—including surgery combined with intraoperative ablative techniques—could be considered for these patients.
- The term carcinomatosis is used in Recommendation 1.2 and Table 5. Given the context of soft tissue sarcoma, sarcomatosis might be a more appropriate term.
- In Recommendation 10.2, the authors state, “For patients with limited liver metastases under systemic therapy, consider focal therapy for dominant-liver lesions either to yield a systemic therapy-free interval or consolidate systemic therapy response.” The term dominant-liver lesions is unclear for me. Are the authors referring to ablating only certain liver metastases while leaving others untreated? Or does this refer to targeting liver metastases selectively in patients who have additional extrahepatic metastases, which remain under systemic treatment? Clarification on this point would be helpful.
Author Response
Reviewer 4
The methodology is appropriate, and the review is based on up-to-date literature. The language is clear, and the article is well-structured and easy to follow.
Author response: Thank you!
- Comment (1) from Reviewer 4: “In section 3.4, point 4.2, the authors recommend, “Hepatic MRI is recommended to diagnose and evaluate lesions smaller than 1 cm and to characterize lesions of uncertain origin.” While this is valid, MRI has been shown to be superior to CT in detecting additional liver lesions, beyond those initially identified by CT. This is particularly relevant for patients being considered for liver resection. Therefore, I believe that MRI should be mandatory in the preoperative workup for all patients undergoing liver metastasectomy, not just for those with lesions smaller than 1 cm.”
Author response: We sincerely appreciate your insightful comment regarding the role of hepatic MRI in the preoperative workup of patients undergoing liver metastasectomy. We completely agree that MRI has demonstrated superiority over CT in detecting additional liver lesions, which is crucial for surgical planning. However, as this consensus aims to provide recommendations with worldwide applicability, we have opted to recommend MRI primarily for the characterization of lesions smaller than 1 cm, acknowledging potential variations in imaging availability and clinical practice across different institutions. To address your concern, we have clarified this point in the manuscript while maintaining the broader applicability of our recommendations. Thank you for your feedback.
- Comment (2) from Reviewer 4: “In Recommendation 4.3, the authors state, “18F-FDG-PET/CT scan should be recommended to rule out extrahepatic metastases in suitable subtypes (e.g., lymph node metastases).” I believe PET/CT should be recommended for all patients considered for liver metastasectomy, not just for specific subtypes. This would help exclude other metastatic sites and detect potential recurrences, thereby optimizing patient selection for surgery.”
Author response: Thank you for your comment. We agree that, while 18F-FDG-PET/CT scan is specifically recommended in certain subtypes (for instance, in NCCN Guidelines for Soft Tissue Sarcoma management), it should also be considered in all patients that are candidates to liver metastasectomy, to exclude metastatic sites and detect potential recurrences. However, it should be noted that we do not believe it should be mandatory, as it may not be available in due time for all patients. We have revised the text in the manuscript, accordingly.
- Comment (3) from Reviewer 4: “In Recommendation 5.5, the authors propose, “Liver surgical resection of OMD of STS should be discouraged if the patient presents with bilobar liver metastatic disease.” However, I do not believe bilobar disease should be considered an absolute contraindication. For instance, in cases where two bilobar metastases can be managed with parenchymal-sparing surgery, resection remains a viable option. Additionally, a multimodal approach - including surgery combined with intraoperative ablative techniques - could be considered for these patients.
Author response: Thank you for pointing this out. There is a typo – we meant “bilobar liver metastatic disease requiring complex hepatectomies. We have revised the text (i.e. Recommendation 5.5) accordingly.
- Comment (4) from Reviewer 4: “The term carcinomatosis is used in Recommendation 1.2 and Table 5. Given the context of soft tissue sarcoma, sarcomatosis might be a more appropriate term.”
Author response: Thank you for pointing this out. We have corrected the term, accordingly.
- Comment (5) from Reviewer 4: “In Recommendation 10.2, the authors state, “For patients with limited liver metastases under systemic therapy, consider focal therapy for dominant-liver lesions either to yield a systemic therapy-free interval or consolidate systemic therapy response.” The term dominant-liver lesions is unclear for me. Are the authors referring to ablating only certain liver metastaseswhile leaving others untreated? Or does this refer to targeting liver metastases selectively in patients who have additional extrahepatic metastases, which remain under systemic treatment? Clarification on this point would be helpful.”
Author response: Thank you for your comment. We rephrased the manuscript for further clarification. The term dominant lesion is very common in focal therapy literature, specially SBRT, SABR and SRS with different definitions depending on its location. Regarding liver lesions, Aisling et al (JCO, 2016) for HCC defined as “the liver lesion at highest risk of growing or rupturing”. It is commonly used to describe a symptomatic lesion among multiple or a lesion that requires focal treatment despite existing others.
Round 2
Reviewer 3 Report
Comments and Suggestions for Authors
Now that the reply is well done, so the manuscript is suitable for publication.
Comments on the Quality of English LanguageFaborable.